# DEEP MULTI-VIEW LEARNING VIA TASK-OPTIMAL CCA

## ABSTRACT

Multi-view learning seeks to form better models by making use of multiple feature sets representing the same samples. Exploiting feature correlations during training can enable better models. The traditional method for computing correlations between feature sets is Canonical Correlation Analysis (CCA), which finds linear projections that maximize correlation between feature vectors, essentially computing a shared embedding for two views of data. More recently, CCA has been used for multi-view discriminative tasks; however, CCA makes no use of class labels. Recent CCA methods have started to address this weakness but are limited in that they do not simultaneously optimize the CCA projection for discrimination and the CCA projection itself, or they are linear only. We address these deficiencies by simultaneously optimizing a CCA-based and a task objective in an end-to-end manner. Together, these two objectives learn a non-linear CCA projection to a shared latent space that is highly correlated and discriminative. Our method shows a significant improvement over previous state-of-the-art (including deep supervised approaches) for cross-view classification (8.5% increase), regularization with a second view during training when only one view is available at test time (2.2-3.2%), and semi-supervised learning (15%) on real data.

## 1 INTRODUCTION

Parallel modalities of data are increasingly common in a variety of applications, including images and text, audio and video, parallel texts of different languages, and a variety of medical imaging and omics modalities for each patient. Each view provides essential information for classification and, when used together, can form a more accurate model. This is especially important for difficult discriminative tasks such as those with a small training set size. Canonical Correlation Analysis (CCA) is the most common method for computing a shared representation from two views of data by computing a space in which they are maximally correlated (Hotelling, 1936; Bie et al., 2005). In this paper we will demonstrate that, through optimizing for both discriminative features and correlation between views, we can improve classification accuracy for three real world scenarios.

CCA is an unsupervised method but has been applied to many discriminative tasks (Kan et al., 2015; Sargin et al., 2007; Arora & Livescu, 2012). While some of the correlated CCA features are useful for discriminative tasks, many represent properties that are of no use for classification and obscure correlated information that is beneficial. This problem is magnified with recent non-linear extensions of CCA that use deep learning to make significant strides in improving correlation (Andrew et al., 2013; Wang et al., 2015a; 2016; Chang et al., 2018) but often at the expense of discriminative capability (cf. §5.1). Therefore, we present Task-Optimal CCA (TOCCA), a new deep learning technique to project the data from two views to a shared space that is also discriminative (Fig. 1).

Implementing a task-optimal variant of CCA required a fundamental change in formulation. We show that the CCA objective can equivalently be expressed as an $\ell_2$ distance minimization in the shared space plus an orthogonality constraint. Orthogonality constraints help regularize neural networks (NNs) (Huang et al., 2018); we present three techniques to accomplish this. While our method is derived from CCA, by manipulating the orthogonality constraints, we obtain deep CCA approaches that compute a shared latent space that is also discriminative.

Our family of solutions for supervised CCA required a crucial and non-trivial change in formulation. We demonstrate the effectiveness and versatility of our model for three different tasks: 1) cross-view classification on a variation of MNIST (LeCun, 1998), 2) regularization when two views are

**Figure 1:** Our goal with Task-Optimal CCA is to compute a shared space that is also discriminative. We do this by using NNs to compute an embedding for each view while simultaneously optimizing for correlation in the embedded space and a task-optimal objective. This setup is beneficial for three scenarios: 1) training a classifier with the embedding from one view and testing with the embedding of the other view (§5.1), 2) when two views are available for training but only one at test time (§5.2), and 3) when both views are used for both training and testing (§5.3). The embeddings for views $X_1$ and $X_2$ are represented by $A_1$ and $A_2$, respectively. A classifier $f(A)$ then predicts the class for each sample. In order to compare with unsupervised variants of CCA, the classifier $f$ may be computed subsequent to the CCA embedding.

available for training but only one at test time on a cancer imaging and genomic data set with only 1,000 samples, and 3) semi-supervised representation learning to improve speech recognition. All experiments showed a significant improvement in accuracy over previous state-of-the-art. In addition, our approach is more robust in the small sample size regime than alternative methods. Overall, our experiments on real data show the effectiveness of our method in learning a shared space that is more discriminative than previous methods for a variety of practical problems.

## 2 RELATED WORK

CCA was initially used for unsupervised data analysis to gain insights into components shared by two sources (Andrew et al., 2013; Wang et al., 2015a; 2016). CCA has also been used to compute a shared latent space for cross-view classification (Kan et al., 2015; Wang et al., 2015a; Chandar et al., 2016; Chang et al., 2018), for representation learning on multiple views that are then joined for prediction (Sargin et al., 2007; Dorfer et al., 2016b), and for classification from a single view when a second view is available during training (Arora & Livescu, 2012). Recent non-linear extensions of CCA implemented via NNs make significant improvements in correlation (Andrew et al., 2013; Wang et al., 2015a; 2016; Chang et al., 2018) but with little focus on discriminative capability.

Most prior work that boosts the discriminative capability of CCA is *linear only* (Lee et al., 2015; Singanamalli et al., 2014; Duan et al., 2016). More recent work using NNs still remains limited in that it optimizes discriminative capability for an intermediate representation rather than the final CCA projection (Dorfer et al., 2016b), or optimizes the CCA objective only during pre-training, not while training the task objective (Dorfer et al., 2018). We advocate to jointly optimize CCA and a discriminative objective by computing the CCA projection within a network layer while applying a task-driven operation such as classification. Experimental results show that our method significantly improves upon previous work (Dorfer et al., 2016b; 2018) due to its focus on both the shared latent space and a task-driven objective. The latter is particularly important on small training set sizes.

While alternative approaches to multi-view learning via CCA exist, they typically focus on a reconstruction objective. That is, they transform the input into a shared space such that the input could be reconstructed – either individually or reconstructing one view from the other. Variations of coupled dictionary learning (Shekhar et al., 2014; Xu et al., 2015; Cha et al., 2015; Bahrampour et al., 2015) and autoencoders (Wang et al., 2015a; Bhatt et al., 2017) have been used in this context. CCA-based objectives, such as the model used in this work, instead learn a transformation to a shared space without the need for reconstructing the input. This task may be easier and sufficient in producing a representation for multi-view classification (Wang et al., 2015a).

## 3 BACKGROUND

We first introduce CCA and present our task-driven approach in §4. Linear and non-linear CCA are unsupervised and find the shared signal between a pair of data sources, by maximizing the sum correlation between corresponding projections. Let $\mathbf{X}_1 \in \mathbb{R}^{d_1 \times n}$ and $\mathbf{X}_2 \in \mathbb{R}^{d_2 \times n}$ be mean-centered input data from two different views with $n$ samples and $d_1$, $d_2$ features, respectively.

**CCA.** The objective is to maximize the correlation between $\mathbf{a}_1 = \mathbf{w}_1^\top \mathbf{X}_1$ and $\mathbf{a}_2 = \mathbf{w}_2^\top \mathbf{X}_2$, where $\mathbf{w}_1$ and $\mathbf{w}_2$ are projection vectors (Hotelling, 1936). The first canonical directions are found via

$$\arg\max_{\mathbf{w}_1, \mathbf{w}_2} \mathrm{corr}\big(\mathbf{w}_1^\top \mathbf{X}_1, \mathbf{w}_2^\top \mathbf{X}_2\big)$$

and subsequent projections are found by maximizing the same correlation but in orthogonal directions. Combining the projection vectors into matrices $\mathbf{W}_1 = [\mathbf{w}_1^{(1)}, \ldots, \mathbf{w}_1^{(k)}]$ and $\mathbf{W}_2 = [\mathbf{w}_2^{(1)}, \ldots, \mathbf{w}_2^{(k)}]$ ($k \leq \min(d_1, d_2)$), CCA can be reformulated as a trace maximization under orthonormality constraints on the projections, i.e.,

$$\arg\max_{\mathbf{W}_1, \mathbf{W}_2} \mathrm{tr}(\mathbf{W}_1^\top \mathbf{\Sigma}_{12} \mathbf{W}_2) \quad \text{s.t. } \mathbf{W}_1^\top \mathbf{\Sigma}_1 \mathbf{W}_1 = \mathbf{W}_2^\top \mathbf{\Sigma}_2 \mathbf{W}_2 = \mathbf{I} \tag{1}$$

for covariance matrices $\mathbf{\Sigma}_1 = X_1 X_1^T$, $\mathbf{\Sigma}_2 = X_2 X_2^T$, and cross-covariance matrix $\mathbf{\Sigma}_{12} = X_1 X_2^T$. Let $\mathbf{T} = \mathbf{\Sigma}_1^{-1/2} \mathbf{\Sigma}_{12} \mathbf{\Sigma}_2^{-1/2}$ and its singular value decomposition (SVD) be $\mathbf{T} = \mathbf{U}_1 \mathrm{diag}(\boldsymbol{\sigma}) \mathbf{U}_2^\top$ with singular values $\boldsymbol{\sigma} = [\sigma_1, \ldots, \sigma_{\min(d_1, d_2)}]$ in descending order. $\mathbf{W}_1$ and $\mathbf{W}_2$ are computed from the top $k$ singular vectors of $\mathbf{T}$ as $\mathbf{W}_1 = \mathbf{\Sigma}_1^{-1/2} \mathbf{U}_1^{(1:k)}$ and $\mathbf{W}_2 = \mathbf{\Sigma}_2^{-1/2} \mathbf{U}_2^{(1:k)}$ where $\mathbf{U}^{(1:k)}$ denotes the $k$ first columns of matrix $\mathbf{U}$. The sum correlation in the projection space is equivalent to

$$\sum_{i=1}^k \mathrm{corr}\big({(\mathbf{w}_1^{(i)})}^\top X_1, {(\mathbf{w}_2^{(i)})}^\top \mathbf{X}_2\big) = \sum_{i=1}^k \sigma_i^2 \ , \tag{2}$$

i.e., the sum of the top $k$ singular values. A regularized variation of CCA (RCCA) ensures that the covariance matrices are positive definite by computing the covariance matrices as $\hat{\mathbf{\Sigma}}_1 = \frac{1}{n-1} \mathbf{X}_1 \mathbf{X}_1^\top + r\mathbf{I}$ and $\hat{\mathbf{\Sigma}}_2 = \frac{1}{n-1} \mathbf{X}_2 \mathbf{X}_2^\top + r\mathbf{I}$, for regularization parameter $r > 0$ and identity matrix $\mathbf{I}$ (Bilenko & Gallant, 2016).

**DCCA.** Deep CCA adds non-linear projections to CCA by non-linearly mapping the input via a multilayer perceptron (MLP). In particular, inputs $\mathbf{X}_1$ and $\mathbf{X}_2$ are mapped via non-linear functions $f_1$ and $f_2$, parameterized by $\theta_1$ and $\theta_2$, resulting in activations $\mathbf{A}_1 = f_1(\mathbf{X}_1; \theta_1)$ and $\mathbf{A}_2 = f_2(\mathbf{X}_2; \theta_2)$ (assumed to be mean centered) (Andrew et al., 2013). When implemented by a NN, $\mathbf{A}_1$ and $\mathbf{A}_2$ are the output activations of the final layer with $d_o$ features. Fig. 2(a) shows the network structure. DCCA optimizes the same objective as CCA (equation 1) but using activations $\mathbf{A}_1$ and $\mathbf{A}_2$. Regularized covariance matrices are computed accordingly and the solution for $\mathbf{W}_1$ and $\mathbf{W}_2$ can be computed using SVD just as with linear CCA. When $k = d_o$ (i.e., the number of CCA components is equal to the number of features in $\mathbf{A}_1$ and $\mathbf{A}_2$), optimizing the sum correlation in the projection space (equation 2) is equivalent to optimizing the following matrix *trace norm objective (TNO)*

$$\mathcal{L}_{\mathrm{TNO}}(\mathbf{A}_1, \mathbf{A}_2) = \|\mathbf{T}\|_{\mathrm{tr}} = \mathrm{tr}\big(\mathbf{T}^\top \mathbf{T}\big)^{1/2} \ ,$$

where $\mathbf{T} = \mathbf{\Sigma}_1^{-1/2} \mathbf{\Sigma}_{12} \mathbf{\Sigma}_2^{-1/2}$ as in CCA (Andrew et al., 2013). DCCA optimizes this objective directly, *without* a need to compute the CCA projection within the network. The TNO is optimized first, followed by a linear CCA operation before downstream tasks like classification are performed. This formulation does not allow for combining directly with a supervised term.

**SoftCCA.** While DCCA enforces orthogonality constraints on projections $\mathbf{W}_1^\top \mathbf{A}_1$ and $\mathbf{W}_2^\top \mathbf{A}_2$, SoftCCA relaxes them using regularization (Chang et al., 2018). Final projection matrices $\mathbf{W}_1$ and $\mathbf{W}_2$ are integrated into $f_1$ and $f_2$ as the top network layer. The trace objective for DCCA in equation 1 can be rewritten as minimizing the $\ell_2$ distance between the projections when each feature in $\mathbf{A}_1$ and $\mathbf{A}_2$ is normalized to a unit variance (Li et al., 2003), leading to[1] $\mathcal{L}_{\ell_2 \text{ dist}}(A_1, A_2) = \|\mathbf{A}_1 - \mathbf{A}_2\|_F^2$ . Regularization in SoftCCA penalizes the off-diagonal elements of the covariance matrix

---

[1] We use this $\ell_2$ distance objective in our formulation.

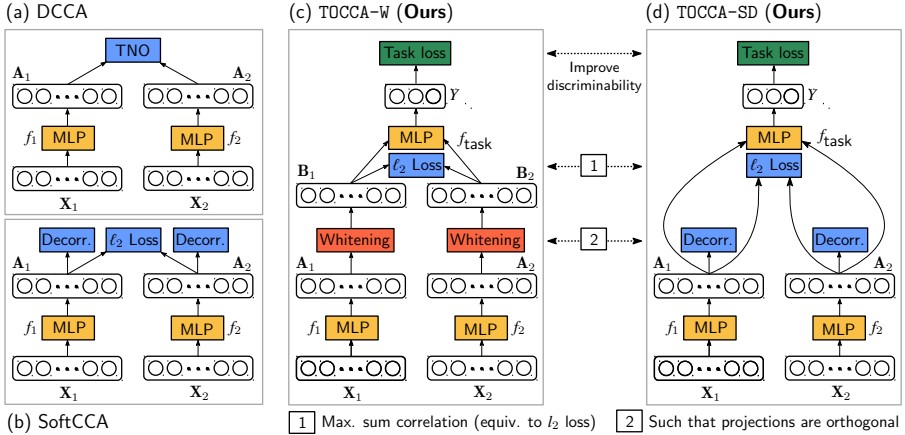

**Figure 2:** Deep CCA architectures: (a) DCCA maximizes the sum correlation in projection space by optimizing an equivalent loss, the trace norm objective (TNO) (Andrew et al., 2013); (b) SoftCCA relaxes the orthogonality constraints by regularizing with soft decorrelation (Decorr) and optimizes the $\ell_2$ distance in the projection space (equivalent to sum correlation with activations normalized to unit variance) (Chang et al., 2018). Our TOCCA methods add a task loss and apply CCA orthogonality constraints by regularizing in two ways: (c) TOCCA-W uses whitening and (d) TOCCA-SD uses Decorr. The third method that we propose, TOCCA-ND, simply removes the Decorr components of TOCCA-SD.

$\boldsymbol{\Sigma}$, using a running average computed over batches as $\hat{\boldsymbol{\Sigma}}$ and a loss of $\mathcal{L}_{\text{Decorr}}(\mathbf{A}) = \sum_{i \neq i}^{d_o} |\hat{\Sigma}_{i,j}|$. Overall, the SoftCCA loss takes the form

$$\mathcal{L}_{\ell_2 \text{ dist}}(\mathbf{A}_1, \mathbf{A}_2) + \lambda\big(\mathcal{L}_{\text{Decorr}}(\mathbf{A}_1) + \mathcal{L}_{\text{Decorr}}(\mathbf{A}_2)\big) \ .$$

**Supervised CCA methods.** CCA, DCCA, and SoftCCA are all unsupervised methods to learn a projection to a shared space in which the data is maximally correlated. Although these methods have shown utility for discriminative tasks, a CCA decomposition may not be optimal for classification because features that are correlated may not be discriminative. Our experiments will show that maximizing the correlation objective too much can degrade performance on discriminative tasks.

CCA has previously been extended to supervised settings in three ways: 1) with methods that are linear only (Singanamalli et al., 2014; Lee et al., 2015; Kan et al., 2015; Duan et al., 2016), 2) by maximizing the total correlation between each view and the training labels in addition to each pair of views (Lee et al., 2015; Singanamalli et al., 2014), and 3) with Linear Discriminant Analysis (LDA)-style approaches to encourage class separation (Kan et al., 2015; Dorfer et al., 2016b; El-madany et al., 2016).[2] LDA approaches to supervision are generative rather than discriminative. Importantly, we will show in §5.3 that encouraging class separation with an LDA-style objective performs significantly inferior to a softmax. Further, Dorfer et al. (2016b) did not apply LDA to the shared space itself but to the NN layer below it, and Elmadany et al. (2016) did not validate the shared space created, only its use in multi-view classification using both views for training and test.

Dorfer et. al's CCA Layer (CCAL) is the closest to our method. It optimizes a task loss operating on a CCA projection; however, the CCA objective itself is only optimized during pre-training, not in an end-to-end manner (Dorfer et al., 2018). Further, their goal is retrieval with a pairwise rank loss, not classification. Instead of computing the CCA projection explicitly within the network, we optimize the non-linear mapping into the shared space *together* with the task objective, requiring a fundamental change in formulation. We optimize for the shared space with the $\ell_2$ distance between activations (similar to SoftCCA) and propose three different ways to apply the orthogonality constraints of CCA.

## 4 TASK-OPTIMAL CCA (TOCCA)

To compute a shared latent space that is also discriminative, we reformulate DCCA to add a task-driven term to the optimization objective. The CCA component finds features that are correlated

---

[2]Gatto & Dos Santos (2017) use a similar technique with LDA but apply it as a convolutional filter on a single view; it is not a multi-view method.

between views, while the task component ensures that they are also discriminative. This model can be used for representation learning on multiple views before joining representations for prediction (Sargin et al., 2007; Dorfer et al., 2016b) and for classification when two views are available for training but only one at test time (Arora & Livescu, 2012). In §5, we demonstrate both use cases on real data. Our methods and related NN models from the literature are summarized in Tab. A2; Fig. 2 shows schematic diagrams.

**Challenges and solutions.** While DCCA optimizes the sum correlation with an equivalent loss function (TNO), the CCA projection itself is computed only *after* optimization. Hence, the projections cannot be used to optimize another task simultaneously. The main challenge in developing a task-optimal form of deep CCA that discriminates based on the CCA projection is in computing this projection within the network – a necessary step to enable simultaneous training of both objectives. We tackle this by focusing on the two components of DCCA: maximizing the sum correlation between activations $\mathbf{A}_1$ and $\mathbf{A}_2$ and enforcing orthonormality constraints within $\mathbf{A}_1$ and $\mathbf{A}_2$. We achieve both by transforming the CCA objective and present three methods that progressively relax the orthogonality constraints.

We further improve upon DCCA by enabling mini-batch computations for improved flexibility and test performance. DCCA was developed for large batches because correlation is not separable across batches. While large batch implementations of stochastic gradient optimization can increase computational efficiency via parallelism, small batch training provides more up-to-date gradient calculations, allowing a wider range of learning rates and improving test accuracy (Masters & Luschi, 2018). We reformulate the correlation objective as the $\ell_2$ distance (following SoftCCA), enabling separability across batches. We ensure a normalization to one via batch normalization without the scale and shift parameters (Ioffe & Szegedy, 2015). Wang et al. (2016) also developed a stochastic mini-batch solution to DCCA but handled the orthonormality constraints in a different way (discussed below).

**Task-driven objective.** First, we apply non-linear functions $f_1$ and $f_2$ with parameters $\theta$ (via MLPs) to each view $\mathbf{X}_1$ and $\mathbf{X}_2$, i.e., $\mathbf{A}_1 = f_1(\mathbf{X}_1; \theta_1)$ and $\mathbf{A}_2 = f_2(\mathbf{X}_2; \theta_2)$. Second, a task-specific function $f_{\text{task}}(\mathbf{A}; \theta_{\text{task}})$ operates on the outputs $\mathbf{A}_1$ and $\mathbf{A}_2$. In particular, $f_1$ and $f_2$ are optimized so that the $\ell_2$ distance between $\mathbf{A}_1$ and $\mathbf{A}_2$ is minimized; therefore, $f_{\text{task}}$ can be trained to operate on both inputs $\mathbf{A}_1$ and $\mathbf{A}_2$. We combine CCA and task-driven objectives as a weighted sum with a hyperparameter for tuning. This model is flexible, in that the task-driven goal can be used for classification (Krizhevsky et al., 2012; Dorfer et al., 2016a), regression (Katzman et al., 2016), clustering (Caron et al., 2018), or any other task. Other prior attempts to integrate a classifier into deep CCA only used LDA (Kan et al., 2015; Dorfer et al., 2016b; Elmadany et al., 2016). See Tab. A2 for an overview.

**Orthogonality constraints.** The remaining complications for mini-batch optimization are the orthogonality constraints, for which we propose three solutions, each handling the orthogonality constraints of CCA in a different way: whitening, soft decorrelation, and no decorrelation.

**1) Whitening (`TOCCA-W`).** CCA applies orthogonality constraints to $\mathbf{A}_1$ and $\mathbf{A}_2$. We accomplish this with a linear whitening transformation that transforms the activations such that their covariance becomes the identity matrix, i.e., features are uncorrelated. Decorrelated Batch Normalization (DBN) has previously been used to regularize deep models by decorrelating features (Huang et al., 2018) and inspired our solution. In particular, we apply a transformation $\mathbf{B} = \mathbf{U}\mathbf{A}$ to make $\mathbf{B}$ orthonormal, i.e., $\mathbf{B}\mathbf{B}^\top = \mathbf{I}$.

We use a Zero-phase Component Analysis (ZCA) whitening transform composed of three steps: rotate the data to decorrelate it, rescale each axis, and rotate back to the original space. Each transformation is learned from the data. Any matrix $\mathbf{U}\epsilon\mathbb{R}^{d_o \times d_o}$ satisfying $\mathbf{U}^\top\mathbf{U} = \mathbf{\Sigma}^{-1}$ whitens the data, where $\mathbf{\Sigma}$ denotes the covariance matrix of $\mathbf{A}$. As $\mathbf{U}$ is only defined up to a rotation, it is not unique. PCA whitening follows the first two steps and uses the eigendecomposition of $\mathbf{\Sigma}$: $\mathbf{U}_{PCA} = \mathbf{\Lambda}^{-1/2}\mathbf{V}^\top$ for $\mathbf{\Lambda} = \text{diag}(\lambda_1, \ldots, \lambda_{d_o})$ and $\mathbf{V} = [\mathbf{v}_1, \ldots, \mathbf{v}_{d_o}]$, where $(\lambda_i, \mathbf{v}_i)$ are the eigenvalue, eigenvector pairs of $\mathbf{\Sigma}$. As PCA whitening suffers from stochastic axis swapping, neurons are not stable between batches (Huang et al., 2018). ZCA whitening uses the transformation $\mathbf{U}_{ZCA} = \mathbf{V}\mathbf{\Lambda}^{-1/2}\mathbf{V}^T$ in which PCA whitening is first applied, followed by a rotation back to the original space. Adding the rotation $\mathbf{V}$ brings the whitened data $\mathbf{B}$ as close as possible to the original data $\mathbf{A}$ (Kessy et al., 2015).

Computation of $\mathbf{U}_{\text{ZCA}}$ is clearly depend on $\mathbf{\Sigma}$. While Huang et al. (2018) used a running average of $\mathbf{U}_{\text{ZCA}}$ over batches, we apply this stochastic approximation to $\mathbf{\Sigma}$ for each view using the update $\mathbf{\Sigma}^{(k)} = \alpha\mathbf{\Sigma}^{(k-1)} + (1-\alpha)\mathbf{\Sigma}^b$ for batch $k$ where $\mathbf{\Sigma}^b$ is the covariance matrix for the current batch and $\alpha \in (0,1)$ is the momentum. We then compute the ZCA transformation from $\mathbf{\Sigma}^{(k)}$ to do whitening as $\mathbf{B} = f_{\text{ZCA}}(\mathbf{A}) = \mathbf{U}_{\text{ZCA}}^{(k)}\mathbf{A}$. At test time, $\mathbf{U}^{(k)}$ from the last training batch is used. Algorithm A1 describes ZCA whitening in greater detail. In summary, TOCCA-W integrates both the correlation and task-driven objectives, with decorrelation performed by whitening, into

$$\mathcal{L}_{\text{task}}(f_{\text{task}}(\mathbf{B}_1), Y) + \mathcal{L}_{\text{task}}(f_{\text{task}}(\mathbf{B}_2), Y) + \lambda\,\mathcal{L}_{\ell_2\text{ dist}}(\mathbf{B}_1, \mathbf{B}_2)\ ,$$

where $\mathbf{B}_1$ and $\mathbf{B}_2$ are whitened outputs of $\mathbf{A}_1$ and $\mathbf{A}_2$, respectively, and $Y$ is the class labels. This is a novel approach to integrating the orthogonality constraints of CCA into a NN as it is the first to use ZCA whitening in this manner. Wang et al. (2016)'s stochastic mini-batch solution to DCCA used nonlinear orthogonal iterations and does not state what type of whitening operation was used.

**2) Soft decorrelation (TOCCA-SD).** While fully independent components may be beneficial in regularizing NNs on some data sets, a softer decorrelation may be more suitable on others. In this second formulation we relax the orthogonality constraints using regularization, following the Decorr loss of SoftCCA (Chang et al., 2018). The loss function for this formulation is

$$\mathcal{L}_{\text{task}}(f_{\text{task}}(\mathbf{A}_1), Y) + \mathcal{L}_{\text{task}}(f_{\text{task}}(\mathbf{A}_2), Y) + \lambda_1\mathcal{L}_{\ell_2\text{ dist}}(\mathbf{A}_1, \mathbf{A}_2) + \lambda_2\big(\mathcal{L}_{\text{Decorr}}(\mathbf{A}_1) + \mathcal{L}_{\text{Decorr}}(\mathbf{A}_2)\big)\ .$$

While this solution is based on SoftCCA, our experiments (§5) will demonstrate that the task component is essential when using the model for classification.

**3) No decorrelation (TOCCA-ND).** When CCA is used in an unsupervised manner, some form of orthogonality constraint or decorrelation is necessary to ensure that $f_1$ and $f_2$ do not simply produce multiple copies of the same feature. While this result could maximize the sum correlation, it is not helpful in capturing useful projections. In the task-driven setting, the discriminative term ensures that the features in $f_1$ and $f_2$ are not replicates of the same information. TOCCA-ND therefore removes the decorrelation term entirely, forming the simpler objective

$$\mathcal{L}_{\text{task}}(f_{\text{task}}(\mathbf{A}_1), Y) + \mathcal{L}_{\text{task}}(f_{\text{task}}(\mathbf{A}_2), Y) + \lambda\mathcal{L}_{\ell_2\text{ dist}}(\mathbf{A}_1, \mathbf{A}_2)\ .$$

These three models allow testing whether whitening or decorrelation benefit a task-driven model.

**Computational complexity.** Due to the eigendecomposition, TOCCA-W has a complexity of $O(d_o^3)$ compared to $O(d_o^2)$ for TOCCA-SD, with respect to output dimension $d_o$. However, $d_o$ is typically small ($\leq 100$) and this extra computation is only performed once per batch. The difference in runtime is less than 6.5% for a batch size of 100 or 9.4% for a batch size of 30 (Tab. A4).

**Summary.** All three variants are motivated by adding a task-driven component to deep CCA. TOCCA-ND is the most relaxed and directly attempts to obtain identical latent representations. Experiments will show that whitening (TOCCA-W) and soft decorrelation (TOCCA-SD) provide a beneficial regularization. Further, since the $\ell_2$ distance that we optimize was shown to be equivalent to the sum correlation (cf. §3 SoftCCA paragraph), all three TOCCA models maintain the goals of CCA, just with different relaxations of the orthogonality constraints. Our method is the first to simultaneously optimize for CCA and a discriminative task with end-to-end training. See Tab. A2 for an overview.

## 5 EXPERIMENTS

We validated our methods on three different data sets: MNIST handwritten digits, the Carolina Breast Cancer Study (CBCS) using imaging and genomic features, and speech data from the Wisconsin X-ray Microbeam Database (XRMB). Our experiments show the utility of our methods for 1) cross-view classification, 2) regularization with a second view during training when only one view is available at test time, and 3) representation learning on multiple views that are joined for prediction.

**Implementation.**[3] Each layer of our network consists of a fully connected layer, followed by a ReLU activation and batch normalization (Ioffe & Szegedy, 2015). Our implementations of DCCA, SoftCCA, and Joint DCCA/DeepLDA (Dorfer et al., 2016b) also use ReLU activation and batch

---

[3]Code is submitted with this paper and will also be available publicly on GitHub after the review period.

normalization. We modified CCAL-$\mathcal{L}_{\text{rank}}$ (Dorfer et al., 2018) to use a softmax function and cross-entropy loss for classification, instead of a pairwise ranking loss for retrieval, referring to this modification as CCAL-$\mathcal{L}_{\text{ce}}$. We used the Nadam optimizer and tuned hyperparameters on a validation set via random search; settings and ranges are specified in Tab. A3. The same hyperparameter tuning procedure was used for our methods and those we compare with. We used Keras with the Theano backend and an Nvidia GeForce GTX 1080 Ti.

The following experiments compare our methods with two linear methods (CCA and RCCA), two unsupervised deep methods (DCCA and SoftCCA), and two supervised deep methods (Joint DCCA/DeepLDA and CCAL-$\mathcal{L}_{\text{ce}}$). Many other variants exist (§3), but the ones we selected are the current state-of-the-art in each of these classes. We did not run a direct comparison with Wang et al. (2015a) as Chang et al. (2018) already showed that SoftCCA is superior. We chose Joint DCCA/DeepLDA to represent supervised LDA-style CCA methods rather than comparing with all methods in this group (Kan et al., 2015; Elmadany et al., 2016)[4].

## 5.1 CROSS-VIEW CLASSIFICATION ON MNIST DIGITS

We formed a multi-view data set from the MNIST handwritten digit data set (LeCun, 1998). Following Andrew et al. (2013), we split each $28 \times 28$ image in half horizontally, creating left and right views that are each $14 \times 28$ pixels. All images were flattened into a vector with 392 features. The full data set consists of 60k training images and 10k test images. We used a random set of up to 50k for training and the remaining training images for validation. We used the full 10k image test set.

In order to validate both the discriminativeness of the embedding and the success in finding a shared space, we studied performance on cross-view classification. We evaluated cross-view classification accuracy by first computing the projection for each view, then we trained a linear SVM on one view's projection, and finally we used the other view's projection at test time. While the task-driven methods presented in this work learn a classifier within the model, this test setup enables a fair comparison with the unsupervised CCA variants and validates the discriminativity of the features learned. It is also the standard method in the literature to test CCA methods for classification. Notably, using the built-in softmax classifier (not shown) performed similarly to the SVM, as much of the power of our methods comes from the representation learning part. We do not compare with a simple supervised NN because this setup does not learn the shared space necessary for cross-view classification. We report results averaged over five randomly selected training/validation sets; the test set always remained the same.

**Correlation vs. classification accuracy** We first demonstrate the importance of adding a task-driven component to DCCA by showing that maximizing the sum correlation between views is not sufficient. Fig. 3 (*left*) shows the sum correlation vs. cross-view classification accuracy across many different hyperparameter settings for DCCA (Andrew et al., 2013), SoftCCA (Chang et al., 2018), and TOCCA. We used 50 components for each; thus, the maximum sum correlation was 50. The sum correlation was measured after applying linear CCA to ensure that components were independent. With DCCA a larger correlation tended to produce a larger classification accuracy, but there was still a large variance in classification accuracy amongst hyperparameter settings that produced a similar sum correlation. For example, with the two farthest right points in the plot (colored red), their classification accuracy differs by 10%, and they are not even the points with the best classification accuracy (colored purple). The pattern is different for SoftCCA. There was an increase in classification accuracy as sum correlation increased but only up to a point. For higher sum correlations, the classification accuracy varied even more from 20% to 80%. Further experiments (not shown) have indicated that when the sole objective is correlation, some of the projection directions are simply not discriminative, particularly when there are a large number of classes. Hence, optimizing for sum correlation alone does not guarantee a discriminative model. TOCCA-W and TOCCA-SD show a much greater classification accuracy across a wide range of correlations and, overall, the best accuracy when correlation is greatest.

**Effect of batch size.** Fig. 3 (*right*) plots the batch size vs. classification accuracy for a training set size of $10,000$. We tested batch sizes from 10 to 10,000; a batch size of 10 or 30 was best for all

---

[4]While Elmadany et al. (2016) ran experiments on MNIST, they used the embeddings from both views for training and test; hence, their results are not directly comparable to our cross-view classification results. When we did test multi-view classification on MNIST, we achieved 98.5% vs. their reported 97.2%.

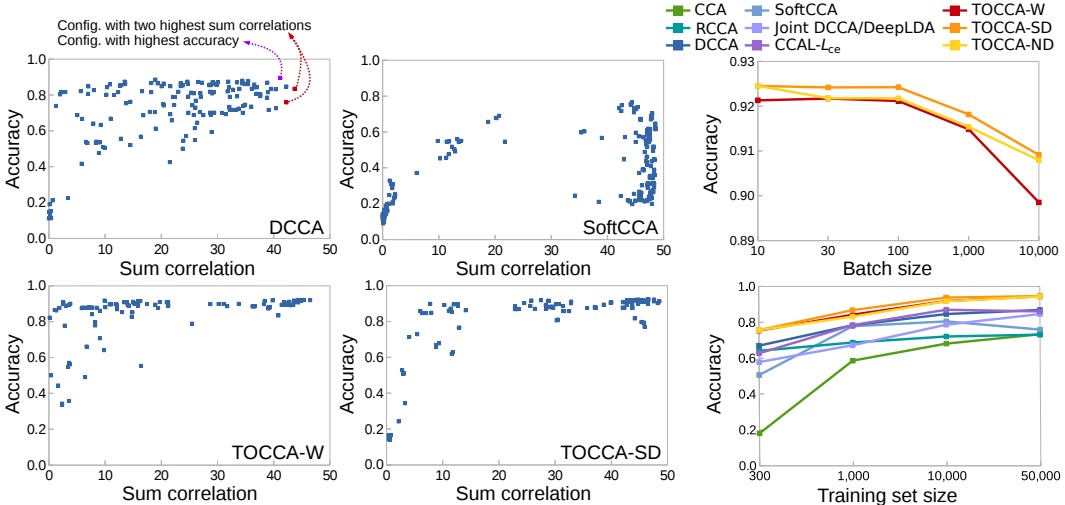

**Figure 3:** *Left*: Sum correlation vs. cross-view classification accuracy (on MNIST) across different hyper-parameter settings on a training set size of 10,000 for DCCA (Andrew et al., 2013), SoftCCA (Chang et al., 2018), TOCCA-W, and TOCCA-SD. For unsupervised methods (DCCA and SoftCCA), large correlations do not necessarily imply good accuracy. *Right*: The effect of batch size on classification accuracy for each TOCCA method on MNIST (training set size of 10,000), and the effect of training set size on classification accuracy for each method. Our TOCCA variants out-performed all others across all training set sizes.

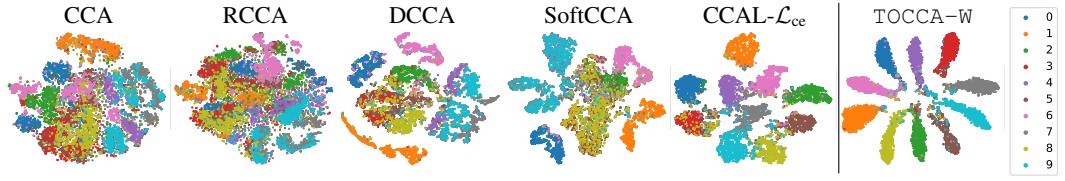

**Figure 4:** *t*-SNE plots for CCA methods on our variation of MNIST. Each method was used to compute projections for the two views (left and right sides of the images) using 10,000 training examples. The plots show a visualization of the projection for the left view with each digit colored differently. TOCCA-SD and TOCCA-ND (not shown) produced similar results to TOCCA-W.

three variations of TOCCA. This is in line with previous work that found the best performance with a batch size between 2 and 32 (Masters & Luschi, 2018). We used a batch size of 32 in the remaining experiments on MNIST.

**Effect of training set size.** We manipulated the training set size in order to study the robustness of our methods. In particular, Fig. 3 (*right*) shows the cross-view classification accuracy for training set sizes from $n = 300$ to $50,000$. While we expected that performance would decrease for smaller training set sizes, some methods were more susceptible to this degradation than others. The classification accuracy with CCA dropped significantly for $n = 300$ and $1,000$, due to overfitting and instability issues related to the covariance and cross-covariance matrices. SoftCCA shows similar behavior (prior work (Chang et al., 2018) on this method did not test such small training set sizes).

Across all training set sizes, our TOCCA variations consistently exhibited good performance, e.g., increasing classification accuracy from 78.3% to 86.7% for $n = 1,000$ and from 86.1% to 94.6% for $n = 50,000$ with TOCCA-SD. Increases in accuracy over TOCCA-ND were small, indicating that the different decorrelation schemes have only a small effect on this data set; the task-driven component is the main reason for the success of our method. In particular, the classification accuracy with $n = 1,000$ did better than the unsupervised DCCA method on $n = 10,000$. Further, TOCCA with $n = 300$ did better than linear methods on $n = 50,000$, clearly showing the benefits of the proposed formulation. We also examined the CCA projections qualitatively via a 2D *t*-SNE embedding (Van Der Maaten & Hinton, 2008). Fig. 4 shows the CCA projection of the left view for each method. As expected, the task-driven variant produced more clearly separated classes.

**Table 1:** Classification accuracy for different methods of predicting Basal genomic subtype from images or grade from gene expression. Linear SVM and DNN were trained on a single view, while all other methods were trained with both views. By regularizing with the second view during training, all TOCCA variants improved classification accuracy. The standard error is in parentheses.

| Method | Training data | Test data | Task | Accuracy | Method | Training data | Test data | Task | Accuracy |
|---|---|---|---|---|---|---|---|---|---|
| Linear SVM | Image only | Image | Basal | 0.777 (0.003) | Linear SVM | GE only | GE | Grade | 0.832 (0.012) |
| NN | Image only | Image | Basal | 0.808 (0.006) | NN | GE only | GE | Grade | 0.830 (0.012) |
| CCAL-$\mathcal{L}_{ce}$ | Image+GE | Image | Basal | 0.807 (0.008) | CCAL-$\mathcal{L}_{ce}$ | GE+image | GE | Grade | 0.804 (0.022) |
| TOCCA-W | Image+GE | Image | Basal | **0.830 (0.006)** | TOCCA-W | GE+image | GE | Grade | **0.862 (0.013)** |
| TOCCA-SD | Image+GE | Image | Basal | 0.818 (0.006) | TOCCA-SD | GE+image | GE | Grade | 0.856 (0.011) |
| TOCCA-ND | Image+GE | Image | Basal | 0.816 (0.004) | TOCCA-ND | GE+image | GE | Grade | 0.856 (0.011) |

## 5.2 REGULARIZATION FOR CANCER CLASSIFICATION

In this experiment, we address the following question: Given two views available for training but only one at test time, does the additional view help to regularize the model?

We study this question using 1,003 patient samples with image and genomic data from CBCS[5] (Troester et al., 2018). Images consisted of four cores per patient from a tissue microarray that was stained with hematoxylin and eosin. Image features were extracted using a VGG16 backbone (Simonyan & Zisserman, 2015), pre-trained on ImageNet, by taking the mean of the 512D output of the fourth set of conv. layers across the tissue region and further averaging across all core images for the same patient. For gene expression (GE), we used the set of 50 genes in the PAM50 array (Parker et al., 2009). The data set was randomly split into half for training and one quarter for validation/testing; we report the mean over eight cross-validation runs. Classification tasks included predicting 1) Basal vs. non-Basal genomic subtype using images, which is typically done from GE, and 2) predicting grade 1 vs. 3 from GE, typically done from images. This is not a multi-task classification setup; it is a means for one view to stabilize the representation of the other. The first task is also a valuable clinical use case. Genomic analysis is expensive and not routinely performed, while histologic imaging is standard practice by pathologists for detecting cancer and assessing its aggressiveness. In working with our clinical collaborators, our goal has been to predict tumor subtypes from images - something that is too complex for pathologists. We hope that this will one day make tumor subtypes accessible to more patients and improve treatment decisions. This experiment demonstrates that the second view of data can help regularize during training even if it is not available for test patients.

We tested different classifier training methods when only one view was available at test time: a) a linear SVM trained on one view, b) a deep NN trained on one view using the same architecture as the lower layers of TOCCA, c) CCAL-$\mathcal{L}_{ce}$ trained on both views, d) TOCCA trained on both views. Tab. 1 lists the classification accuracy for each method and task. When predicting genomic subtype Basal from images, all our methods showed an improvement in classification accuracy; the best result was with TOCCA-W, which produced a 2.2% improvement. For predicting grade from GE, all our methods again improved the accuracy – by up to 3.2% with TOCCA-W. These results show that having additional information during training can boost performance at test time. Notably, this experiment used a static set of pre-trained VGG16 image features in order to assess the utility of the method. The network itself could be fine-tuned end-to-end with our TOCCA model, providing an easy opportunity for data augmentation and likely further improvements in classification accuracy.

## 5.3 SEMI-SUPERVISED LEARNING FOR SPEECH RECOGNITION

Our final experiments use speech data from XRMB, consisting of simultaneously recorded acoustic and articulatory measurements. Prior work has shown that CCA-based algorithms can improve phonetic recognition (Wang et al., 2015b;a; 2016; Dorfer et al., 2016b). The 45 speakers were split into 35 for training, 2 for validation, and 8 for testing – a total of 1,429,236 samples for training, 85,297 for validation, and 111,314 for testing.[6] The acoustic features are 112D and the articulatory ones are 273D. We removed the per-speaker mean & variance for both views. Samples are annotated with one of 38 phonetic labels.

---

[5] http://cbcs.web.unc.edu/for-researchers/
[6] http://ttic.uchicago.edu/~klivescu/XRMB_data/full/README

Our task on this data set was representation learning for multi-view prediction – that is, using both views of data to learn a shared discriminative representation. We trained each model using both views and their labels. To test each CCA model, we followed prior work and concatenated the original input features from both views with the projections from both views. Due to the large training set size, we used a Linear Discriminant Analysis (LDA) classifier for efficiency. The same construction was used at test time. This setup was used to assess whether a task-optimal DCCA model can improve discriminative power. We tested TOCCA with a task-driven loss of LDA (Dorfer et al., 2016a) or softmax to demonstrate the flexibility of our model.

**Table 5:** XRMB classification results.

| Method | Task | Accuracy |
|---|---|---|
| Baseline | - | 0.591 |
| CCA | - | 0.589 |
| RCCA | - | 0.588 |
| DCCA | - | 0.620 |
| SoftCCA | - | 0.635 |
| Joint DCCA/DeepLDA | LDA | 0.633 |
| CCAL-$\mathcal{L}_{ce}$ | Softmax | 0.642 |
| TOCCA-W | LDA | 0.710 |
| TOCCA-SD | LDA | 0.677 |
| TOCCA-ND | LDA | 0.677 |
| TOCCA-W | Softmax | **0.795** |
| TOCCA-SD | Softmax | 0.785 |
| TOCCA-ND | Softmax | 0.785 |

We compared the discriminability of a variety of methods to learn a shared latent representation. Tab. 5 lists the classification results with a baseline that used only the original input features for LDA. Although deep methods, i.e., DCCA and SoftCCA, improved upon the linear methods, all TOCCA variations significantly outperformed previous state-of-the-art techniques. Using softmax consistently beat LDA by a large margin. TOCCA-SD and TOCCA-ND produced equivalent results as a weight of 0 on the decorrelation term performed best. However, TOCCA-W showed the best result with an improvement of 15% over the best alternative method.

**Table 6:** Semi-supervised classification results on XRMB using TOCCA-W.

| Labeled data | Accuracy |
|---|---|
| 100% | 0.795 |
| 30% | 0.762 |
| 10% | 0.745 |
| 3% | 0.684 |
| 1% | 0.637 |

TOCCA can also be used in a *semi-supervised* manner when labels are available for only some samples. Tab. 6 lists the results for TOCCA-W in this setting. With 0% labeled data, the result would be similar to DCCA. Notably, a large improvement over the unsupervised results in Tab. 5 is seen even with labels for only 10% of the training samples.

# 6 DISCUSSION

We proposed a method to find a shared latent space that is also discriminative by adding a task-driven component to deep CCA while enabling end-to-end training. This required a fundamental change in formulation because Deep CCA *does not* compute the embeddings directly as it optimizes an equivalent objective; therefore, we could not simply add an additional term. Instead, we found an alternative formulation by replacing the CCA projection with $\ell_2$ distance minimization and orthogonality constraints on the activations, and we implemented this in three different ways. TOCCA-W or TOCCA-SD performed the best, dependent on the data set – both of which include some means of decorrelation to provide a regularizing effect to the model and thereby outperforming TOCCA-ND.

TOCCA showed large improvements over state-of-the-art in cross-view classification accuracy on MNIST and significantly increased robustness to a small training set size. On CBCS, TOCCA provided a regularizing effect when both views were available for training but only one at test time. TOCCA also produced a large increase over state-of-the-art for multi-view representation learning on a much larger data set, XRMB. On this data set we also demonstrated a semi-supervised approach to get a large increase in classification accuracy with only a small proportion of the labels. Using a similar technique, our method could be applied when some samples are missing a second view.

Classification tasks using a softmax operation or LDA were explored in this work; however, the formulation presented can also be used with other tasks such as regression or clustering. Another possible avenue for future work entails extracting components shared by both views as well as individual components. This approach has been developed for dictionary learning (Lock et al., 2013; Ray et al., 2014; Feng et al., 2018) but could be extended to deep CCA-based methods. Finally, we have yet to apply data augmentation to the proposed framework; this could provide a significant benefit for small training sets.

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

# A    APPENDIX

This appendix includes additional details on our `TOCCA` algorithm and experiments, including 1) a comparison of our formulation with other related CCA approaches, 2) pseudocode for the ZCA whitening algorithm used by `TOCCA-W`, 3) details on hyperparameter selection, and 4) training runtime experiments.

## A.1    COMPARISON OF `TOCCA` WITH RELATED ALGORITHMS

Our `TOCCA` methods finds a shared latent space that is also discriminative by changing the CCA formulation in order to add a task-driven component. Tab. A2 compares our three `TOCCA` formulations with other related methods (discussed in §3). CCA is the baseline linear method with a goal of maximizing the correlation between a set of orthogonal linear projections on two views of data. DCCA and SoftCCA are unsupervised deep methods. DCCA optimizes an equivalent objective to CCA but uses non-linear projections implemented with a NN; however, the projections are not computed in the network, only after optimization is complete. SoftCCA changes the correlation objective to, equivalently, minimize the $\ell_2$ distance between projections and relaxes the orthogonality constraints by using regularization. CCAL-$\mathcal{L}_{\text{rank}}$ does compute the CCA projections in the network but does not optimize the final NN for correlation; it instead focuses on a pairwise ranking loss for use in retrieval. Our family of `TOCCA` methods were detailed in §4. In this supervised formulation, we use the same $\ell_2$ distance as SoftCCA and simultaneously optimize a task-driven objective. We handle the orthogonality constraints in three different ways: with whitening (`TOCCA-W`), with regularization (`TOCCA-SD`) as was used in SoftCCA, and with no explicit decorrelation (`TOCCA-ND`).

**Table A2:** A comparison of our proposed task-optimal deep CCA methods with other related ones from the literature: DCCA (Andrew et al., 2013), SoftCCA (Chang et al., 2018), CCAL-$\mathcal{L}_{\text{rank}}$ (Dorfer et al., 2018). CCAL-$\mathcal{L}_{\text{rank}}$ uses a pairwise ranking loss with cosine similarity to identify matching and non-matching samples for image retrieval – not classification. $A_1$ and $A_2$ are mean centered outputs from two feed-forward networks. $\Sigma = A^T A$ is computed from a single (large) batch (used in DCCA); $\hat{\Sigma}$ is computed as a running mean over batches (for all other methods). $f_{\text{task}}(A; \theta_{\text{task}})$ is a task-specific function with parameters $\theta_{\text{task}}$, e.g., a softmax operation for classification.

| Method | Objective | |
|---|---|---|
| CCA | $-\text{tr}(W_1^T \Sigma_{12} W_2)$ | s.t. $W_1^T \Sigma_1 W_1 = W_2^T \Sigma_2 W^2 = I$ |
| DCCA | $-\|\Sigma_1^{-1/2} \Sigma_{12} \Sigma_2^{-1/2}\|_{\text{tr}}$ | where $\|T\|_{\text{tr}} = \text{tr}(T^T T)^{1/2}$ (TNO, equivalent to CCA objective) |
| | | CCA$(W_1^T A_1, W_2^T A_2)$ computed after optimization complete |
| SoftCCA | $\mathcal{L}_{\ell_2 \text{ dist}}(A_1, A_2) + \lambda \left( \mathcal{L}_{\text{Decorr}}(A_1) + \mathcal{L}_{\text{Decorr}}(A_2) \right)$ | |
| CCAL-$\mathcal{L}_{\text{rank}}$ | $\mathcal{L}_{\text{rank}}(B_1, B_2)$ | where $B_1, B_2 = $ CCA$(A_1, A_2)$, $\mathcal{L}_{\text{rank}}$ is pairwise ranking loss |
| `TOCCA-W` | Task$(B_1, B_2, Y) + \lambda\,\mathcal{L}_{\ell_2 \text{ dist}}(B_1, B_2)$ | where $B_1 = U_1 A_1, B_2 = U_2 A_2$ s.t. $B_1^T B_1 = B_2^T B_2 = I$ |
| `TOCCA-SD` | Task$(A_1, A_2, Y) + \lambda_1 \mathcal{L}_{\ell_2 \text{ dist}}(A_1, A_2) + \lambda_2 \left( \mathcal{L}_{\text{Decorr}}(A_1) + \mathcal{L}_{\text{Decorr}}(A_2) \right)$ | Whitening |
| `TOCCA-ND` | Task$(A_1, A_2, Y) + \lambda\,\mathcal{L}_{\ell_2 \text{ dist}}(A_1, A_2)$ | |

| **Loss functions** | | |
|---|---|---|
| $\ell_2$ dist | $\mathcal{L}_{\ell_2 \text{ dist}}(A_1, A_2) = \|A_1 - A_2\|_F^2$ | |
| Decorr | $\mathcal{L}_{\text{Decorr}}(A) = \sum_{i \neq j} |\hat{\Sigma}_{i,j}|$ | where $\hat{\Sigma}$ is running mean across batches of $\Sigma = A^T A$ |
| Task | Task$(A_1, A_2, Y) = \mathcal{L}_{\text{task}}(f_{\text{task}}(A_1; \theta_{\text{task}}), Y) + \mathcal{L}_{\text{task}}(f_{\text{task}}(A_2; \theta_{\text{task}}), Y)$ | where $\mathcal{L}_{task}$ can be cross-entropy or any other task-driven loss |

## A.2    ALGORITHM FOR WHITENING

`TOCCA-W` uses whitening to achieve orthogonality (see §4 for details). The goal is to transform the activations such that their covariance becomes the identity matrix. We use ZCA whitening which first applies PCA whitening to decorrelate the data and rescale each axis, followed by a rotation back to the original space. The final rotation reduces the stochastic axis swapping problems of PCA whitening (Huang et al., 2018). Pseudocode for ZCA whitening is shown in Algorithm A1.

---

**Algorithm A1** Whitening layer for orthogonality.

---

**Input:** activations $A \epsilon \mathbb{R}^{d_o \times n}$
**Hyperparameters:** batch size $m$, momentum $\alpha$
**Parameters of layer:** mean $\mu$, covariance $\Sigma$
**if** training **then**
    $\mu \leftarrow \alpha\mu + (1-\alpha)\frac{1}{m}A\,1_{n \times 1}$ {Update mean}
    $\bar{A} = A - \mu$ {Mean center data}
    $\Sigma \leftarrow \alpha\Sigma + (1-\alpha)\frac{1}{m-1}\bar{A}_1\bar{A}_2^T$ {Update covariance}
    $\hat{\Sigma} \leftarrow \Sigma + \epsilon I$ {Add $\epsilon I$ for numerical stability}
    $\Lambda, V \leftarrow \text{eig}(\hat{\Sigma})$ {Compute eigendecomposition}
    $U \leftarrow V\Lambda^{-1/2}V^T$ {Compute transformation matrix}
**else**
    $\bar{A} \leftarrow A - \mu$ {Mean center data}
**end if**
$B \leftarrow U\bar{A}$ {Apply ZCA whitening transform}
**return** $B$

---

## A.3 IMPLEMENTATION DETAILS: HYPERPARAMETERS

A random search over hyperparameters was used to train our methods and those that we compare with. The hyperparameter settings and ranges for each data set are provided in Tab. A3. Random search in these intervals was performed 100 times for MNIST and CBCS. Fewer tries were done for XRMB because of the much greater runtime on this large data set. A larger batch size was used for XRMB to improve runtime. The hyperparameter ranges were initially set as an educated guess and, in some cases, were widened for a particular data set (for all methods) after observing results.

**Table A3:** Hyperparameter settings and search ranges for the experiments on each data set.

| Hyperparameter | MNIST | CBCS | XRMB |
|---|---|---|---|
| Hidden layers | 4 | [0,4] | 4 |
| Hidden layer size | 500 | 200 | 1,000 |
| Output layer size | 50 | 50 | 112 |
| Loss function weight $\lambda$ | $[10^0, 10^{-4}]$ | $[10^1, 10^{-5}]$ | $[10^1, 10^{-5}]$ |
| Momentum $\alpha$ | 0.99 | 0.99 | 0.99 |
| Weight decay | $[10^{-3}, 10^{-6}], 0$ | $[10^{-2}, 10^{-5}], 0$ | $[10^{-3}, 10^{-7}], 0$ |
| Soft decorrelation regularizer | $[10^0, 10^{-5}]$ | $[10^0, 10^{-5}]$ | $[10^0, 10^{-5}]$ |
| Batch size | 32 | 100 | 50,000 |
| Learning rate | $[10^{-2}, 10^{-4}]$ | $[10^{-1}, 10^{-3}]$ | $[10^0, 10^{-4}]$ |
| Epochs | 200 | 400 | 100 |

## A.4 RUNTIME EXPERIMENTS

The computational complexity of `TOCCA-W` is greater than that of `TOCCA-SD` due to the eigende-composition operation (see the end of §4); however, this extra computation is only carried out once per batch. A runtime comparison of the two methods on all three data sets is provided in Tab. A4. The difference in runtime was less than 6.5% for a batch size of 100 or 9.4% for a batch size of 30.

**Table A4:** Training runtime for each data set.

| Data set | Batch size | Epochs | TOCCA-W | TOCCA-SD |
|---|---|---|---|---|
| MNIST | 100 | 200 | 488 s | 418 s |
| MNIST | 30 | 200 | 1071 s | 1036 s |
| CBCS | 100 | 400 | 103 s | 104 s |
| XRMB | 50,000 | 100 | 3056 s | 3446 s |

