# OpenReview forum: "Deep Multi-View Learning via Task-Optimal CCA"
_ICLR.cc/2020/Conference — Reject_

### Official Review · AnonReviewer2 · 2019-10-22
**Official Blind Review #2**

**Rating:** 3

**Review:**

Originality:

CCA is a generative model that learns a shared subspace based on  two (or multi) views of the data. Being generative, it might not have strong discriminative power for some downstream classification tasks. Previous approaches to infuse discriminative power into the shared subspace estimated by CCA are linear. So, this paper proposes to learn 1) non-linear 2) discriminative subspaces for CCA. The paper accomplishes this by simply adding a task specific term to the optimization objective of DeepCCA (Andrew et. al. 2013), which involves just adding a task-specific MLP on top and minimizing the associated loss-function.


1). The novelty of the proposed approach is limited. It just adds an extra term (extra neural network layer) with a corresponding weighting hyperparameter to the objective function of a previous method (DeepCCA) without much motivation.


2). The experimental setup and results are sound but some of the tasks seem contrived to show the improved performance of TOCCA methods. For instance, in the cross-view MNIST classification the authors use only projection from one view at training time and use the other view at test-time. What's the motivation for this setup? Why not split the data into train and test set by splitting observations, then train on both the views at train time and test on the held-out observations at test-time? I hope I am not missing something.

3). Similarly, for the "Regularization for Cancer Classification" task, it's assumed that only one view is available at test time. Why is that? What are the real-world examples of such setups?


Quality:

The paper is technically sound, though it is a trivial extension of a previous method. The experimental setup is somewhat contrived to show the superiority of the proposed method.


Clarity:

The paper is well organized and is well written in general. The supplementary material contains more results and code will be available after the review period.


Significance:

The paper solves an important problem by infusing discriminative power into generative subspaces learned by CCA but the results are not that important in my eyes. Since the empirical setup is a little contrived it is hard to even know whether a simple two-step approach that first estimates CCA subspace and then uses those projections in a SVM or MLP would perform comparable or better if given a fair-chance to compete.


**Experience Assessment:**

I have published in this field for several years.

**Review Assessment: Checking Correctness Of Derivations And Theory:**

I assessed the sensibility of the derivations and theory.

**Review Assessment: Checking Correctness Of Experiments:**

I carefully checked the experiments.

**Review Assessment: Thoroughness In Paper Reading:**

I read the paper thoroughly.

---

> ### Author Response · Authors · 2019-11-07
> **Response**
>
> Thank you for your feedback!
>
> We provide a response to each of your points below:
>
> Originality 1: We proposed a deep, discriminative variant of CCA to enable end-to-end training for classification or other tasks. Integrating supervision into existing deep CCA methods required a fundamental change in formulation. Deep CCA does not compute the embeddings directly as it optimizes an equivalent objective; therefore, we could not simply add an additional term. Instead, we found an alternative formulation for maximizing the correlation and three different ways to handle the orthogonality constraints of CCA. We contributed these 3 CCA-inspired solutions that progressively relax the orthogonality constraints of CCA. None of these 3 approaches correspond to anything in the literature. Further, our approach gives significant accuracy improvements on three different types of analysis tasks: 8.5% improvement for cross-view classification on MNIST, 2.2-3.2% for regularization with a second view on CBCS, and 15% for multi-view learning on XRMB.
>
> Originality 2: We tested 3 distinct use cases: C1) cross-view classification, C2) multi-view regularization during training, and C3) semi-supervised representation learning for multi-view prediction.
>
> In our cross-view classification experiment on MNIST (Sec. 4.1), we trained an SVM on the embedding from the first view and tested using the embedding from the second view. This toy setup validates both the discriminativeness of the embedding and the success in finding a shared space. Our results showed clear deficiencies in previous deep CCA methods and demonstrated that ours is much more robust to small training sets.
>
> Real world use cases are best represented by C2 and C3. C3 studies multi-view classification for phonetic recognition on speech data using XRMB. Both views were used for training and at test time. This experiment demonstrated our TOCCA methods for learning a better representation for classification and seems to be the test case that you suggested in your comment about our MNIST experiment. We found a 15% increase in classification with TOCCA vs. the best deep supervised CCA-based methods that we compared with.
>
> Originality 3: Our other real world test case (and the one that motivates our work) is the cancer classification experiment for C2 (Sec. 4.2). We demonstrated that TOCCA can regularize a model when two views are available for training but only one at test time. This example uses a breast cancer data set with imaging and genomic data and was motivated by our clinical collaborators. Specifically, all patients undergoing a tumor biopsy or resection will be screened by a pathologist, i.e., the pathologist will inspect the tissue sample under a microscope to detect cancer and assess its aggressiveness. This analysis is comparatively cheap and part of the established workflow. Genomic analysis, which may be performed to inform treatment decisions based on genomic subtype (see https://www.breastcancer.org/symptoms/testing/types/prosigna), is expensive. Hence, having an image-based screening method to determine who would benefit from further genomic analysis could be much more cost effective and enable this technology to benefit more patients. Our results demonstrated that having the extra genomic view during training can improve upon an image-only model. Our clinical collaborators on this project are interested in this exact use case.
>
> Note that this setup was specifically chosen based on strong practical needs of our medical collaborators where the assumption is that only one of the modalities will be available during test time. We agree that, if one had both modalities at test time, this experimental setup would not be considered optimal. Our setup was not chosen to highlight the performance of TOCCA, instead TOCCA was designed (and experimentally tested) to fulfill the requirements for a real clinical problem. We are happy to clarify this in the manuscript.
>
> Quality: The three use cases that we studied are not contrived; each is suited to a different application. This diversity of experiments demonstrated the versatility of our approach. Note also (and see also above) that we do not only add a term, but propose and explore a family of different relaxations. Even if our approach is considered simple, it shows strong improvements over state-of-the-art methods on multiple datasets and for multiple tasks. This makes TOCCA, in our opinion, a highly practical and useful approach.
>
> Clarity: Code will be on GitHub after the review period; however, it was also submitted anonymously with this paper (stated on pg. 6, footnote 3).

---

> > ### Author Response · Authors · 2019-11-07
> > **Response (continued)**
> >
> > (continued)
> >
> > Significance: Regarding a comparison using the CCA projection and an SVM, we did include such an experiment on the MNIST data set (the CCA line in Fig. 2). For XRMB we tested CCA followed by LDA (Tbl. 4). Both of these tests were significantly inferior to our TOCCA methods (21.3% improvement for MNIST and 20.6% for XRMB). As explained in detail above, we do not consider our experimental setup contrived. For example, the breast cancer analysis (and its experimental setup) is motivated by a clear clinical need for a disease likely to affect 1 in 8 women in the US in their lifetime (https://www.breastcancer.org/symptoms/understand_bc/statistics).

---

### Official Review · AnonReviewer1 · 2019-10-22
**Official Blind Review #1**

**Rating:** 3

**Review:**

This is a interesting paper on an important topic, but it was a main weakness: it assumes that the reader is deeply familiar with the CCA. In order to make the paper more accessible to a general audience, the authors should:
1) have at least one sentence in the abstract that explains in layman terms why is CCA important and how it works ("multi-view learning" does not suffice); given that you have the term "multi-view learning" in the title, you should explain what it is and how it can benefit from CCA
2)  re-organize the current intro, which reads more like related work, into a more traditional format
     - one intuitive paragraph on what is multi-view learning (MVL), what is CCA, how does CCA help MVL
     - one intuitive paragraph on an illustrative example on how MVL & CCA help solving a problem
     - one intuitive paragraph on how the proposed approach works
     - one paragraph summarizing the main findings/results
3) ideally, add a section with an illustrative running example, which would have a huge impact on the paper's readability (far more than, say, than the current Appendix)

**Experience Assessment:**

I do not know much about this area.

**Review Assessment: Checking Correctness Of Derivations And Theory:**

I did not assess the derivations or theory.

**Review Assessment: Checking Correctness Of Experiments:**

I assessed the sensibility of the experiments.

**Review Assessment: Thoroughness In Paper Reading:**

I read the paper at least twice and used my best judgement in assessing the paper.

---

> ### Author Response · Authors · 2019-11-07
> **Response**
>
> Thank you for your feedback!  We will get to work on revising the abstract and intro to provide a clearer introduction to multi-view learning and CCA. We apologize if it was not easy to follow.
>
> Specifically, we will make the following changes based on your comments:
>
> 1) We will revise the first sentence of our abstract to read “Multi-view learning seeks to form better models by making use of multiple feature sets representing the same samples.  Exploiting feature correlations during training can enable better models.  The traditional method for computing correlations between feature sets is Canonical Correlation Analysis (CCA), which finds linear projections that maximize correlation between feature vectors, essentially computing a shared embedding for two views of data. More recently, CCA has been used for multi-view discriminative tasks; however, CCA makes no use of class labels....”
>
> 2) We will reorganize the manuscript as suggested and add the desired paragraphs.
>
> 3) We are not sure we entirely understand your suggestion. Fig. 1 covers the architecture differences between the models, so is it just a higher level illustration that you feel would help?  For example, we could add a simple graphical illustration motivating the creation of a shared embedding that is also discriminative and how that is formed with CCA/DCCA then improved with TOCCA.
>
> Would these clarifications address your concerns? We will work on them now and post them as they are completed.

---

> > ### Author Response · Authors · 2019-11-15
> > **Follow-up response**
> >
> > We have now updated our paper to include the abstract and intro changes mentioned in our previous comment.  Figure 1 was also added to better illustrate the problem and applications.  We hope that these changes alleviate your concerns.

---

### Official Review · AnonReviewer3 · 2019-10-23
**Official Blind Review #3**

**Rating:** 6

**Review:**

This paper addresses the problem of jointly performing CCA with task labeling. The problem is timely and important as it is challenging to perform CCA jointly with the task classification (see below) and hence previous work typically perform this in a pipeline - that is, first projecting the data using a pre-trained CCA and then training a task classifier using the projected representation. As the authors note, this may be problematic as CCA may delete important information that is relevant for the classification, if training is not done jointly.

As the authors note, the main challenge in developing a task-optimal form of deep CCA that discriminates based on the CCA projection is in computing this projection within the network. To deal with this, the authors propose approximations for two steps: maximizing the sum correlation between activations A1 and A2 and enforcing orthonormality constraints within A1 and A2. Particularly, the authors present three methods that progressively relax the orthogonality constraints. Also, because correlation is not separable across batches, SGD is not possible for deep CCA training and the authors deal with this too (although this was dealt with in previous work on deep CCA as well).

This is an empirical paper in the sense that no guarantees are provided for the proposed techniques. The experiments are thorough, convincing and the span a range of applications. The results demonstrate the value of the proposed approximations, and I hence recommend a weak accept of the paper.

**Experience Assessment:**

I have published one or two papers in this area.

**Review Assessment: Checking Correctness Of Derivations And Theory:**

I assessed the sensibility of the derivations and theory.

**Review Assessment: Checking Correctness Of Experiments:**

I assessed the sensibility of the experiments.

**Review Assessment: Thoroughness In Paper Reading:**

I read the paper at least twice and used my best judgement in assessing the paper.

---

> ### Author Response · Authors · 2019-11-07
> **Response**
>
> Thank you for the kind feedback!  Are there any areas in which we could improve our paper in your opinion? We would be happy to do so.

---

### Comment · Area_Chair1 · 2019-11-14
**Reviewers, any comments on the author responses?**

Dear Reviewers, thanks for your thoughtful input on this submission!  The authors have now responded to your comments.  Please be sure to go through their replies and revisions.  If you have additional feedback or questions, it would be great to get them this week while the authors still have the opportunity to respond/revise further.  Thanks!

---

### Decision · Program_Chairs · 2019-12-19

**Decision:**

Reject

**Comment:**

The main contribution of this paper is the training of a supervised model jointly with deep CCA for improving the representations learned in a setting where the training data is multi-view.  The claimed technical contribution is modifications to deep CCA to enable it to play nicely with the minibatch gradient-based training used for the supervised loss.  Pros:  This is an important problem with many applications.  Cons:  The novelty is minimal.  Some previous work has done joint training of supervised models with CCA, and some has addressed training deep CCA in a stochastic setting.  The reviewers (and I) are unconvinced that the differences from previous work are sufficient, and the paper does not carefully compare with the previous work.  The contribution to the tasks may be quite significant, however, so the paper may fit in well in an application-oriented conference/journal.